# The response of terrestrial ecosystem carbon cycling under different aerosol-based radiation management geoengineering

Hanna Lee[1], Helene Muri[2], Altug Ekici[1,3,4], Jerry Tjiputra[1], and Jörg Schwinger[1]

[1]NORCE Norwegian Research Institute, Bjerknes Centre for Climate Research, Bergen, Norway
[2]Industrial Ecology Programme, Department of Energy and Process Engineering, Norwegian University of Science and Technology, Trondheim, Norway
[3]Current address: Climate and Environmental Physics, Physics Institute, University of Bern, Bern, Switzerland
[4]Current address: Oeschger Centre for Climate Change Research, University of Bern, Bern, Switzerland

**Correspondence:** Hanna Lee (hanna.lee@norceresearch.no)

**Abstract.** Geoengineering has been discussed as a potential option to offset the global impacts of anthropogenic climate change, and at the same time by reaching global temperature targets of the Paris Agreement. Before any implementation of geoengineering, however, the complex natural responses and consequences of such methods should be fully understood to avoid any unexpected and potentially degrading impacts. Here we assess the changes in ecosystem carbon exchange and storage

among different terrestrial biomes under three aerosol based radiation management methods with the baseline of RCP8.5 using an Earth System Model (NorESM1-ME). All three methods used in this study (stratospheric aerosol injection, marine sky brightening, cirrus cloud thinning) target the global mean radiation balance at the top of the atmosphere to reach that of the RCP4.5 scenario. The three RM methods investigated in this study show vastly different precipitation patterns especially in the tropical forest biome. Precipitation differences from the three RM methods result large variability in global vegetation

carbon uptake and storage. Our findings show that there are unforeseen regional consequences under geoengineering and these consequences should be taken into account in future climate policies as they have substantial impact on terrestrial ecosystems. Although, changes in temperature and precipitation play a large role in vegetation carbon uptake and storage, our results show that $CO_2$ fertilization also plays a considerable role. We find that the effects of geoengineering on vegetation carbon storage was much smaller than the effects of mitigation under RCP4.5 scenario (e.g. afforestation in the tropics). Our results emphasizes

the importance of considering multiple combined effects and responses of land biomes while reaching the global temperature targets of the Paris Agreement.

## 1 Introduction

The Paris Agreement, adopted under the Convention of the Parties of United Nations Framework Convention on Climate Change (UNFCCC) in 2015, aims to limit the temperature increase to 2°C, and strive for 1.5°C above pre-industrial levels

(UNFCCC, 2015). This temperature target is very ambitious considering the rate of current warming as such goals require not only strong mitigation efforts (e.g., Rogelj et al., 2016, 2018; van Vuuren et al., 2018), and also negative emission technologies or carbon dioxide removal (CDR) are likely needed (IPCC, 2018). Geoengineering has been discussed as a potential option to

offset the global impacts of anthropogenic climate change, and at the same time help reach those global temperature targets. The complex natural responses and consequences of such methods, however, should be fully understood before implementation
of geoengineering to avoid any unexpected and potentially degrading impacts.

By definition, geoengineering is a deliberate attempt to modify the climate system on a sufficiently large scale to alleviate the impacts of climate change (Crutzen, 2006). Two broad categories of geoengineering, which have been discussed persistently in the Fifth Assessment Report of the Intergovernmental Panel on Climate Change (IPCC, 2013), are CDR and Solar Radiation Management (SRM). The CDR methods aim at capturing $CO_2$ from the atmosphere and storing it in reservoirs, where it
stays isolated from the atmosphere for a significant period of time. This could be done in a number of different ways, from afforestation to direct air capture of $CO_2$ with long-term geological storage (Lawrence et al., 2018). The SRM methods, on the other hand, aim at modifying the atmospheric radiative budgets by reducing the amount of solar radiation reaching the Earth's surface to alleviate anthropogenic global warming. We hence refer to these methods as radiation management (RM) in this study, following Schäfer et al. (2015).

Due to the long thermal inertia in the climate system and limitations in maximum removal rate of $CO_2$, CDR would likely require longer time to lower global temperatures (Zickfeld et al., 2017) compared to RM methods. On the other hand, several proposed RM methods could stabilize or even reduce global temperature within a few years (Lawrence et al., 2018). The benefits of RM methods may not only be in reducing the current rate of increase in atmospheric temperatures, but also in mitigating climate extremes likely caused by warming (Irvine et al., 2019). Despite this encouraging potential, studies have
shown numerous undesirable climatic and biophysical side effects of RM particularly related to sudden termination of RM (e.g., Keller et al., 2014; Lauvset et al., 2017; Lee et al., 2019; Robock et al., 2009; Tjiputra et al., 2016). These studies point out that upon sudden termination of RM, the climate system will return to its "unmitigated" state within a few decades. This may lead to very large rates of change in the climatic state, unless there is a solution to reduce atmospheric $CO_2$ concentrations. Nevertheless, our understanding on how RM influence vegetation carbon (C) dynamics at regional scales remains limited, with
only a few studies published, with focus on single or simplistic RM methods (Dagon and Schrag, 2019; Muri et al., 2014, 2018; Naik et al., 2003; Tjiputra et al., 2016; Xia et al., 2016; Yang et al., 2020; Zhang et al., 2019).

In this study, we assess the response of different terrestrial biomes in their ecosystem C exchange and storage under three different RM methods using an Earth System Model. There are a number of different methods studied within RM, including the aerosol injection based ones used in this study. The three RM methods considered in this study are stratospheric aerosol injec-
tion (SAI), marine sky brightening (MSB), and cirrus cloud thinning (CCT). The mechanism how different methods stabilize the climate are quite different, where SAI and MSB regulate shortwave radiation and CCT modifies terrestrial radiation. Among the three, the most studied is SAI (e.g., Robock, 2016; Tilmes et al., 2015; Tjiputra et al., 2016), which involves increasing the backscatter of solar radiation to space by introducing an reflective aerosol layer in the stratosphere. Bright and reflective aerosols also form the foundation of another method; namely MCB (Ahlm et al., 2017; Alterskjær et al., 2013; Latham et al.,
2012). The principle here is to inject aerosols such as sea salt into low cloud layers over the tropical oceans, to make these more effective at reflecting incoming surface, and hence avoid some surface warming. If such spraying is done outside of the typical cloud deck areas, the brightness of the aerosols themselves may also cool the climate (Ahlm et al., 2017). Hence the term 'ma-

rine sky brightening (MSB)' has been used in the literature (Schäfer et al., 2015; Ahlm et al., 2017), since the sky and not just the clouds are brightened. Furthermore, there is a less studied method referred to as CCT (Gasparini et al., 2020; Mitchell and Finnegan, 2009; Kristjánsson et al., 2015), which aims to cool by letting more longwave radiation escape to space by removing or thinning out high level ice clouds (cirrus clouds). This could also be done by seeding with aerosols. Since thinning of cirrus clouds primarily would act on the longwave range of the spectrum, as opposed to the other two aforementioned method, put together we refer to them as RM rather than the commonly used term SRM to consider cirrus cloud thinning.

The modeling study by Muri et al. (2018) demonstrates that all three of these methods could potentially stabilize atmospheric temperature and reduce net radiative forcing on climate; however, side effects may exist as these methods may alter atmospheric circulation and precipitation patterns. Geoengineering Model Intercomparison Project (GeoMIP, Kravitz et al. (2015)) studies show that there is substantial regional climate variation in response to different methods, scenarios, and models (e.g., Stjern et al., 2018; Wei et al., 2018; Yu et al., 2015). As a result, different terrestrial ecosystems exhibit varying patterns in vegetation production (net primary productivity, NPP). Analyses in vegetation responses show that global mean and high latitude NPP have different patterns (Jones et al., 2013; Lee et al., 2019). This is likely due to different RM methods resulting in different patterns of precipitation in particular. In addition to temperature and precipitation, different biomes are limited by different environmental factors, such as growing season length, dry season length, availability of sunlight for photosynthesis, and soil fertility.

This led us to investigate the following questions: 1) What are the key factors affecting future vegetation under different RM applications? 2) If there are regional differences in the environmental change under RM application, which terrestrial biomes are affected the most in ecosystem C uptake and storage? 3) What is the impact of geoengineering termination on vegetation and terrestrial C storage? 4) What are the effects of RM applications in global vegetation compared to lower emissions and mitigation scenario (RCP4.5)?

### 1.1 Model description (NorESM)

We conducted three different aerosol-based geoengineering experiments using the fully coupled NorESM1-ME, where we investigated the impacts of idealized scenarios of aerosol-based geoengineering under a high-$CO_2$ RCP8.5 and the target temperature scenario RCP4.5 future scenarios. NorESM1-ME is based on the Community Earth System Model (CESM; Gent et al., 2011). Some of the key differences in NorESM1-ME from CESM are: (1) a more sophisticated tropospheric chemistry-aerosol-cloud scheme (Kirkevag et al., 2013), (2) a different ocean circulation model based on the Miami Isopycnic Coordinate Ocean Model (MICOM) with extensive modifications (Bentsen et al., 2013), and (3) the ocean biogeochemical model, which originated from the Hamburg Oceanic Carbon Cycle (HAMOCC) model (Tjiputra et al., 2013). Both the land and atmospheric components have a horizontal resolution of 1.9° latitude × 2.5° longitude, with 26 vertical levels in the atmosphere, whereas the ocean model employs a displaced pole grid with a nominal ~1° resolution with 53 isopycnal layers.

The land component of NorESM1-ME is CLM4 (Lawrence et al., 2011). The land C cycle module in CLM4 includes carbon-nitrogen (CN) coupling that is prognostic in CN as well as vegetation phenology; thus, in addition to temperature and precipitation, plant photosynthesis is also limited by the nitrogen (N) availability (Thornton et al., 2009). The CLM4 has

separate state variables for C and N and these are followed through separately in leaf, live stem, dead stem, live coarse root, dead coarse root, and fine root pools. There are two corresponding storage pools representing short-term and long-term storage of non-structural carbohydrates and labile N. Sources and sinks of mineral N are implemented in the form of atmospheric deposition, biological N fixation, denitrification, leaching, and losses due to fire events. The CLM4 photosynthesis uses both direct and diffuse radiation for sunlit leaves and only diffuse radiation for shaded leaves (Bonan et al., 2011). The plant functional types (PFTs) and land cover change distribution in CLM4 is prescribed and updated annually according to the Coupled Model Intercomparison Project phase 5 (CMIP5) global land use and land cover change data set (Lawrence et al., 2011, 2012). The transient PFT and land cover fields take into account historical and future climate change under RCP8.5 scenario (1850-2100), which were implemented using the harmonized land use change scenarios and Integrated Assessment Model, respectively. Details on PFT, terrestrial C and N cycling, and land cover implementation in the CLM4 model are described in Lawrence et al. (2011). For this study, NorESM is run with fully interactive prognostic C cycle (i.e. in emission driven mode).

## 1.2   Aerosol-based geoengineering experiments

Two Two of the RM methods used in this study aim to reduce the amount of solar radiation reaching the surface to alleviate global warming through spraying of aerosols into the atmosphere: SAI (e.g., Crutzen, 2006; Robock, 2016) and MSB (Ahlm et al., 2017; Latham, 1990). Another technique aims at increasing the amount of outgoing thermal radiation to space by reducing the cover of high level ice clouds: CCT (Storelvmo et al., 2013). Increasing application of RM was used to lower the total radiative forcing in the RCP8.5 baseline simulation down to a temperature level corresponding to RCP4.5, as described in Muri et al. (2018) and similar to the G6sulfur experiment of GeoMIP (Kravitz et al., 2015). The RM is started in year 2020 on the background of the RCP8.5 scenario and continued until the end of the century. The mean of three ensemble members were used for each case. In year 2101 the RM was ended. One ensemble member was extended for another 50 years for each case, such that effects of sudden termination of large-scale RM may be assessed.

The aerosol-based RM experiments were implemented as follows:

### 1.2.1   Stratospheric aerosol injections (RCP8.5 + SAI)

Since there is no interactive stratospheric aerosol scheme in NorESM1-ME, stratospheric aerosol properties were prescribed based on the approach of Tilmes et al. (2015), although different reference cases are used. In simulations with the ECHAM5 model, sulfur dioxide was released at ~2 km altitude (60 hPa) in a grid box at the equator. The interactive aerosol microphysics module within the general circulation model of ECHAM5 (Niemeier et al., 2011) calculated the resulting distribution of sulfate aerosols in the stratosphere. The aerosol optical depth and distribution represented by the zonal aerosol extinction, single-scattering albedo, and asymmetry factors, were implemented in NorESM1-ME, and are described in more detail in Niemeier and Timmreck (2015). A number of test runs were performed to establish how much aerosols were needed to offset the anthropogenic radiative forcing between RCP8.5 and RCP4.5. The resulting aerosol layer correspond to equivalent emissions of 5 Tg(S) yr$^{-1}$ in 2050, 10 Tg(S) yr$^{-1}$ in 2075 and as much as 20 Tg(S) yr$^{-1}$ in 2100.

### 1.2.2 Marine sky brightening (RCP8.5 + MSB)

The sea salt emissions parameterisation in NorESM1-ME is coupled to the cloud droplet number concentrations. In this way, the emissions of sea salt may interact with cloud processes, including brightening effects. Emissions of sea salt aerosols were uniformly increased between latitudes of ±45°. This follows the approach of Alterskjær et al. (2013) and the emissions are increased over a wider latitude band to achieve an effective radiative forcing of -4 W m$^{-2}$ more readily. The medium sized aerosol bin has been found to be most efficient at brightening clouds in NorESM (Alterskjær and Kristjánsson, 2013). Aerosol emissions were hence increased for the accumulation mode size, with a dry number modal radius of 0.13 $\mu m$, geometric standard deviation of 1.59, corresponding to a dry effective radius of 0.22 $\mu m$. Sea salt emission increases were of the order of 460 Tg yr$^{-1}$ at the end of the century.

### 1.2.3 Cirrus cloud thinning (RCP8.5 + CCT)

With regards to cirrus cloud thinning, the Muri et al. (2014) method was used. The fall speed of all ice crystals at temperatures below -38°C was increased. This is a typical temperature for homogeneous freezing to start occurring. The coverage of ice clouds in the CMIP5 ensemble was assessed by Li et al. (2012), and NorESM was found to perform reasonable compared to satellite observations and indeed one of the better-performing models. The terminal velocity of ice was increased by a factor of 10 by 2100, i.e. within the observational range (Gasparini et al., 2017; Mitchell, 1996).

### 1.2.4 Analysis of biomes

We follow the definition of different land biomes as in Tjiputra et al. (2016), where plant functional types (PFT) in the CLM4 that represent certain biomes are merged together (e.g. Boreal forest biome includes boreal needleleaf evergreen tree, boreal needleleaf deciduous tree, boreal broadleaf deciduous tree, and boreal broadleaf deciduous shrub PFTs in the CLM4). The biomes are static by taking a 20 year mean (1981-2000) of PFT distribution from the surface dataset. See Supplementary Information 1 for overall distribution of the biomes used in this study. We note that projected land use change characteristics are very different in RCP8.5 and RCP4.5 (Hurtt et al., 2011). While there is an increase in cropland and grassland (driven by food demand of an increasing population) at the expense of forested land in RCP8.5, there is an increasing area of forest due to assumed reforestation programs in the mitigation scenario RCP4.5 (van Vuuren et al., 2011).

## 2 Results and Discussion

### 2.1 Global scale responses under RM applications

The three RM methods alter the direct visible radiation (DVR) and diffuse visible radiation (FVR) in different directions, with little impact on the level of atmospheric $CO_2$ concentrations (Figure 1). The differences in direct and diffuse radiation is attributed to how the radiation management methods are implemented, where each differs in affecting longwave and shortwave radiation (Muri et al., 2018). Regardless of the methodological differences, all RM methods are able to reduce the net radiation

at the top of the atmosphere and the global mean air temperatures close to the RCP4.5 level as expected. Global land surface air temperature (TSA) increases at a slower rate until the end of the 21st century under all three RM scenarios compared to the baseline RCP8.5 scenario, where there is approximately 2.3°C difference between RM and non-RM world at year 2100. Under CCT application, the increase in global precipitation is somewhat higher than RCP8.5 as explained in Muri et al. (2018). CCT keeps the level of precipitation close to RCP8.5 until the year 2100 due to amplified hydrological cycle from increased latent

heat flux (Kristjánsson et al., 2015). CCT has been shown to lead to an increase in precipitation in previous studies (Jackson et al., 2016; Kristjánsson et al., 2015; Muri et al., 2018), where the radiative cooling of the troposphere increases the latent heat flux at the surface and hence the precipitation rates. SAI shows a reduced rate of increase in global precipitation similar to RCP4.5. Under MSB application, the rate of global precipitation increase falls in between the SAI and CCT.

There is a large overall increase in global mean NPP until the end of the 21st century in the RCP8.5 scenario and under the

165 three RM scenarios (Figure 1), whereas only a small increase in NPP is simulated under the RCP4.5 scenario. At the same time, there is a large increase in the rate of soil organic matter decomposition (heterotrophic respiration: HR) in the RCP8.5 based experiments. Relatively small NPP difference are observed between RCP8.5 and the RM simulations compared to the RCP4.5 scenario. This illustrates that the $CO_2$ fertilization effect is much larger in regulating NPP than the effects of temperature and precipitation, as the level of temperature and (in the case of SAI and MSB) precipitation are similar between the RCP4.5

scenario and the three RM methods on a global scale.

## 2.2 Regional differences in temperature and precipitation

There is no discernable spatial patterns shown in the changes in direct and diffuse radiation except that changes in the CCT application are more concentrated in the tropics. The SAI method shows considerabely increased diffuse radiation throughout the global land areas compared to the baseline RCP8.5 scenario (Supplementary Information 2). While TSA exhibit similar

patterns across different RM applications, the precipitation patterns are more variable over space acorss different RM methods applied. The global spatial patterns of precipitation towards the end of the century (mean of 2070-2100) show (Figure 2) that CCT generally increases precipitation in the tropics and Mediterranean region relative to RCP8.5. In particular, MSB tends to increase precipitation over extratropical land more than SAI due to the regional application of the forcing (Alterskjær et al., 2012). The spatial patterns of precipitation change in MSB mostly follows that of CCT, but the manitude of change is smaller.

On the other hand, SAI shows overall decreases in precipitation particularly in the tropics relative to RCP8.5. All three methods show decrease in precipitation in east Asia region.

The differences in temperature and precipitation across the tree RM methods in different land biomes of the world show that there is no noticeable difference in mean annual temperature across the three different RM methods (Figure 3). The cooling imbalance across the three RM forcings exist, where the tropics tend to cool more than high latitudes and is more pronounced in

the ocean than on land, with a stronger southern hemispheric cooling for CCT (Muri et al., 2018). We show that precipitation patterns vary across the three methods in different biomes. In all biomes, SAI application results in the largest decrease in precipitation followed by MSB relative to RCP8.5 scenario. Under CCT application, precipitation even increases beyond the RCP8.5 level. The precipitation differences across the three methods are large, particularly in the tropics and the mid-latitudes,

where CCT application results in higher precipitation rates than the two other methods. The difference in precipitation becomes amplified over time until the end of the 21st century. According to Muri et al. (2018), shortwave radiation based geoengineering methods exhibit strong reductions in global precipitation levels relative to RCP8.5, but also relative to RCP4.5. CCT leads to a slight increase in global precipitation even over RCP8.5 levels, however, land precipitation patterns in different biomes vary. Aggregated over all biomes, precipitation changes are much smaller than over the total (ocean+land) area. Particularly, precipitation is not reduced much below RCP4.5 levels for SAI and MSB as in Muri et al. (2018) (compare their Figure 2 with Figure 1 in this study).

## 2.3 Biome specific C uptake and release rate

The spatial patterns and the magnitude of NPP change under the three methods show distinct difference. There are common spatial patterns in NPP decrease at north western part of Amazonia, equatorial Africa, and eastern Asia in the three RM experiments (Figure 2). But overall, the large increase in NPP at Europe and equatorial South America particularly CCT experiment compensates the decreases elsewhere, hence creating a general lack of deviation as a whole from RCP8.5 scenario (see Figure 1). It is clear from the comparison shown in precipitation (Figure 2) that the NPP changes are most correlated by the spatial changes in precipitation.

Under CCT application, there is a strong increase in NPP in the tropics and the Mediterranean region, but decrease in East Asia. MSB does not show noticeable change except increased NPP in Eastern Amazonia. The spatial pattern of NPP in MSB is similar to CCT but the magnitude is smaller in MSB. There is a strong decrease in NPP under SAI application, particularly in the tropics. These overall patterns follow similar spatial patterns as the precipitation and are highly correlated as expected (Supplementary Information 2). The differences in NPP is dominated largely by three biomes: tropical forest, grass-shrubland, and temperate forest (Figure 4). NPP and HR in MSB and SAI simulations negatively deviate from the RCP8.5 simulation, whereas in CCT both remain at a similar level to the RCP8.5 in the tropical forest, grass-shrubland, and temperate forest likely due to increased precipitation level in these biomes. But since temperature is a stronger regulator of NPP and HR in high latitude biomes, CCT simulations also exhibit decreased NPP and HR there compared to the RCP8.5 scenario. Additionally, we do not observe any noticeable changes in seasonality for NPP and LAI between RM methods and RCP8.5 scenario (Supplementary Information 3 and 4) as seen in Dagon and Schrag (2019). Although there is spatial patterns in precipitation patters, there is no change in seasonality between the three RM methods and the baseline RCP8.5 scenario.

Overall, the varying precipitation patterns may be the strongest driver of the responses of global scale C uptake and release. Changes in diffuse radiation is found to affect photosynthesis (Keppel-Aleks and Washenfelder, 2016) as diffuse radiation can be more efficient in photosynthesis (e.g., Gu et al., 2002). Under these assumptions, increase in diffuse radiation and decrease in direct radiation under SAI is expected to increase plant photosynthesis (Mercado et al., 2009; Xia et al., 2016). Increases in diffuse radiation is known to positively affect photosynthesis upto a threshold of the ratio between diffuse and total radiation at around 0.4-0.45 (Knohl and Baldocchi, 2008; Mercado et al., 2009). Across the three RM methods, this ratio ranges from 0.29 (CCT) to 0.4 (SAI) at the end of the RM application in year 2100. The responses of NPP under changes in diffuse radiation om different RM applications exhibited in our study suggest that changes in diffuse radiation may not be the largest driver

of NPP change at the global level as much as temperature and precipitation. Under the coupled framework of ESM, it is very difficult to decompose the direct single effects of climatic factors due to interactions (Zhang et al., 2019) and separate simulations are necessary to directly quantify this. The N limitation implemented in the CLM4 has shown to limit C uptake by 74% relative to the C only model (Thornton et al., 2007), but CLM4 still exhibits NPP biases in the tropics (Lawrence et al., 2011). It is important to note that despite the strong $CO_2$ fertilization as well as increase in diffuse radiation, NPP in some parts of the tropics decrease under the SAI application likely due to the strong decrease in precipitation (Figure 2, Supplementary Information 2 and 3).

## 2.4 Biome specific C storage

Vegetation C storage in different biomes illustrate that global vegetation C storage changes are dominated by the responses in tropical forest biome (Figure 5). Under the baseline RCP8.5 scenario, global vegetation C storage decreases due to reduced tropical and temperate forest and grass-shrubland area as part of the land use change scenario used in the RCP8.5 scenario (Riahi et al., 2011). Compared to the baseline RCP8.5 scenario, vegetation C in Arctic tundra, boreal forest, and tropical forest biomes are affected the most under RM application. In Arctic tundra and boreal forest biomes, all three RM scenarios result in a slightly reduced accumulation in vegetation C compared to RCP8.5 scenario likely due to decreased temperature exhibiting the temperature limitation in high latitude biomes. In tropical forest, SAI application reduces vegetation C storage relative to the RCP8.5 scenario, but CCT application slightly increases C storage due to increased precipitation (Figure 2). The magnitude of change in global vegetation C at the end of the century due to application of different RM methods are up to 10 PgC. On the other hand, the magnitude of vegetation C reduced due to the different underlying land use change scenarios in RCP4.5 and RCP8.5 is up to 100 PgC (Figure 5, Supplementary Information 6). These differences are attributed to increased forest and grassland area as part of the RCP4.5 scenario (Thomson et al., 2011). This highlights that large scale changes in vegetation C storage depends much more on anthropogenic land use change than on additional perturbations caused by application of RM in our simulations.

In tropical forest, the differences in vegetation C storage appeared to be correlated to precipitation patterns, where decrease and increase in precipitation in the three different methods regulate vegetation C storage. Differences in vegetation and soil C storage in temperate zone (temperate forest and grass-shrubland), however, did not always correspond directly to varying precipitation patterns. For instance, approximately 100-120 mm difference in mean annual precipitation shown in temperate forest and grass-shrubland biomes between SAI and CCT methods do not portray into differences in vegetation C storage (Figure 3, Supplementary Information 5).

## 2.5 Effects of RM termination

Upon sudden termination of RM application, the level of radiation, temperature, and precipitation quickly converge to the baseline RCP8.5 scenario (Figures 1, 3, 4). Note that the temperature does not increase to exactly the same level as the RCP8.5 scenario, which has been observed in previous studies and is due to the thermal inertia of ocean heat uptake (Tjiputra et al., 2016). As the temperature and precipitation patterns converge towards the RCP8.5 scenario, NPP also becomes similar to the

RCP8.5 scenario (Figure 1). The soil C storage decreases as RM is terminated and towards the end of the simulation in year 2150, soil C storage in all three RM methods are at a similar level (Supplementary Information 5), but the magnitude is still higher than under the RCP8.5 scenario by 10 PgC globally. The likely accumulation of soil C under RM application may be viewed as one of the positive effects of geoengineering, which was supported by a recent multimodel comparison study

(Yang et al., 2020). Globally, land C accumulation associated with RM would remain on land for at least 50 years following termination (Muri et al., 2018). Although the termination effects seem catastrophic due to its rapidity in particular, some studies suggest that realistically the most extreme cases would be unlikely as termination could be avoided by geopolitical agreement once deployed (Parker and Irvine, 2018).

## 2.6 Implications and limitations

Reduced atmospheric temperature and precipitation under RM have large effects on vegetation C storage compared to the baseline scenario, RCP8.5. Under the RCP4.5 scenario, the rate of C uptake denoted as NPP is slower due to reduced temperature, precipitation, and atmospheric $CO_2$ levels (Figure 1). However, global vegetation C storage is far greater than the RCP8.5 and the three RM simulations, which are based on underlying RCP8.5 scenario assumptions (Figure 5), due to the larger forest and grassland areas in the RCP4.5 scenario (Thomson et al., 2011). As a result, the difference in global vegetation C between the

RCP4.5 scenario and the rest of the RCP8.5 based scenarios is nearly 170 PgC. This strongly suggests that on a global scale, the areal change of vegetation and land surface mangement play very important roles when accounting for the global scale vegetation C storage. We suggest taking this point into account when comparing the different pros and cons of technological applications such as geoengineering and mitigation options such as afforestation.

Our results suggest that even with reduced temperature stress created by RM application, productivity of vegetation in the

275 three most productive biomes on Earth may be reduced due to changing precipitation patterns (particularly SAI). Therefore, considering the changes (i.e., reduction) in precipitation alone, RM may have negative effects on non-irrigated crops or food production globally. Nevertheless, the effect of $CO_2$ fertilization effect in the future are suggested to compensate the deleterious impacts of both RM-induced temperature and precipitation changes (Pongratz et al., 2012; Xia et al., 2016). Although not directly investigated in this study, different RM methods have shown various climate extremes as well as mitigating them,

which will have profound effects on the physiology of vegetation (Aswathy et al., 2015). Indeed, some studies show seasonal variation in temperature under geoengineering (Dagon and Schrag, 2019), although we did not observe this in our study. This is not within the scope of our study, but could be an interesting point to consider in future studies.

We acknowledge that CLM4 has numerous limitations that prevents it from accurately estimating global scale soil C storage, and therefore, we do not make an estimation of global soil C storage. But here, we compare soil C storage under different

methods to understand the factors controlling the difference across the three RM methods. Soil C storage increases under RM application compared to the baseline RCP8.5 scenario (Supplementary Information 5 and 6), because the decrease in temperature slows the rate of soil organic matter decomposition by microorganisms. An increase in total soil C is also simulated under the RCP4.5 scenario (Supplementary Information 6) likely due to the combination between increased vegetation C accumulation and slight reduction in soil respiration. There is an increase in soil C storage under RCP8.5 scenario early 21st

century due to increased NPP, but ultimately soil C decreases quickly due to excellerated soil respiration (Figure 1). In different biomes, temperate forest exhibits the largest difference across the three RM methods, where soil C storage under SAI method is nearly 1.0 PgC higher than CCT at the end of the 21st century. This is likely due to lower precipitation in SAI, which reduced the rate of decomposition.

## 3    Conclusions

We show that the three different RM application mainly differ in the precipitation patterns, which in turn affect differences in global scale NPP. The precipitation differences across the three RM application are the most pronounced in the tropics and mid-latitudes, where SAI application results in the largest decrease in precipitation followed by MSB, and CCT relative to RCP8.5 scenario. Tropical forest shows largest variability in NPP and vegetation C storage as the precipitation patterns vary the most across the three methods in the tropics compared to other biomes. Ultimately, all three RM applications investigated in this study reduced the surface temperature to the level of RCP4.5 scenario with vegetation C uptake and storage being affected due to different temperature and precipitation patterns created by the different RM methods. Our results illustrate that there are regional differences in the biogeochemical cycles under application of large scale RM and suggest that such effects should be taken into consideration in future shaping of climate policies. Although changes in temperature and precipitation plays a large role in vegetation C storage capacity, $CO_2$ fertilization plays a considerable role in terrestrial C dynamics that can overshadow the effects of temperature and precipitation. Furthermore, changes in vegetation C storage under large-scale RM application was much smaller than that exhibited under RCP4.5 scenario, which uses climate mitigation efforts by afforestation in the tropics. Hence, it would be important to consider the multiple combined effects and responses of land biomes when applying different strategies to reach the global temperature targets of the Paris Agreement.

*Data availability.* The model simulations used in this study are archived and available at the Norwegian Research Data Archive server (https://doi.org/10.11582/2019.00007).

*Author contributions.* HM and JT received funding; HM and JT designed and conducted simulations; HL and AE analyzed the data; HL, HM, and JS wrote the manuscript; all authors contributed to editing of the manuscript

*Competing interests.* There are no competing interests present.

*Disclaimer.* TEXT

*Acknowledgements.* This research was supported by the Research Council of Norway projects EXPECT (229760/E10), EVA (229771), and HiddenCosts (268243); Bjerknes Centre for Climate Research strategic project SKD-LOES. The simulations were performed on resources provided by UNINETT Sigma2 - the National Infrastructure for High Performance Computing and Data Storage in Norway, accounts nn9182k, nn9448k, NS2345K, and NS9033K.

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

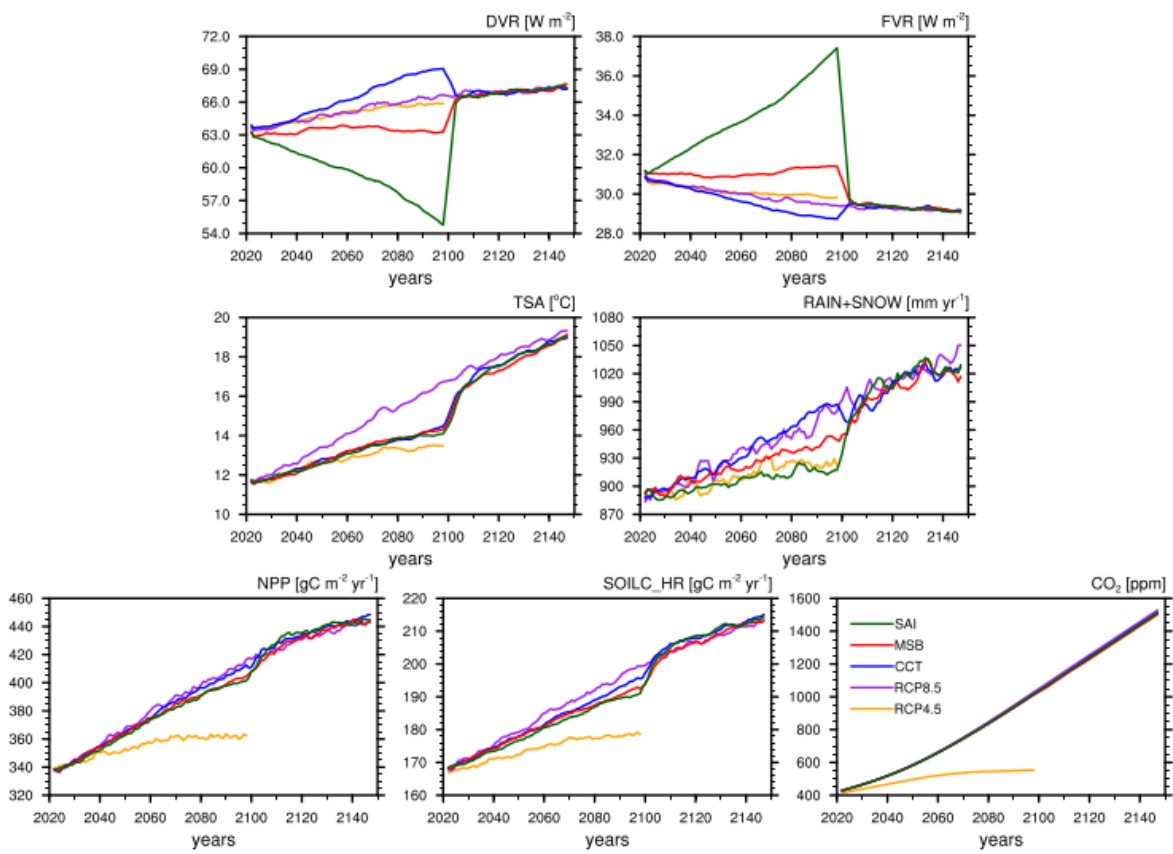

**Figure 1.** Time-series of land surface direct visible solar radiation (DVR), diffuse visible solar radiation (FVR), 2m air temperature (TSA), precipitation (RAIN+SNOW), net primary production (NPP), heterotrophic soil respiration (SOILC_HR), and atmospheric $CO_2$ from RCP4.5, RCP8.5, CCT, MSB, and SAI experiments. The values are spatial means over the land area between $60°$S and $70°$N latitudes.

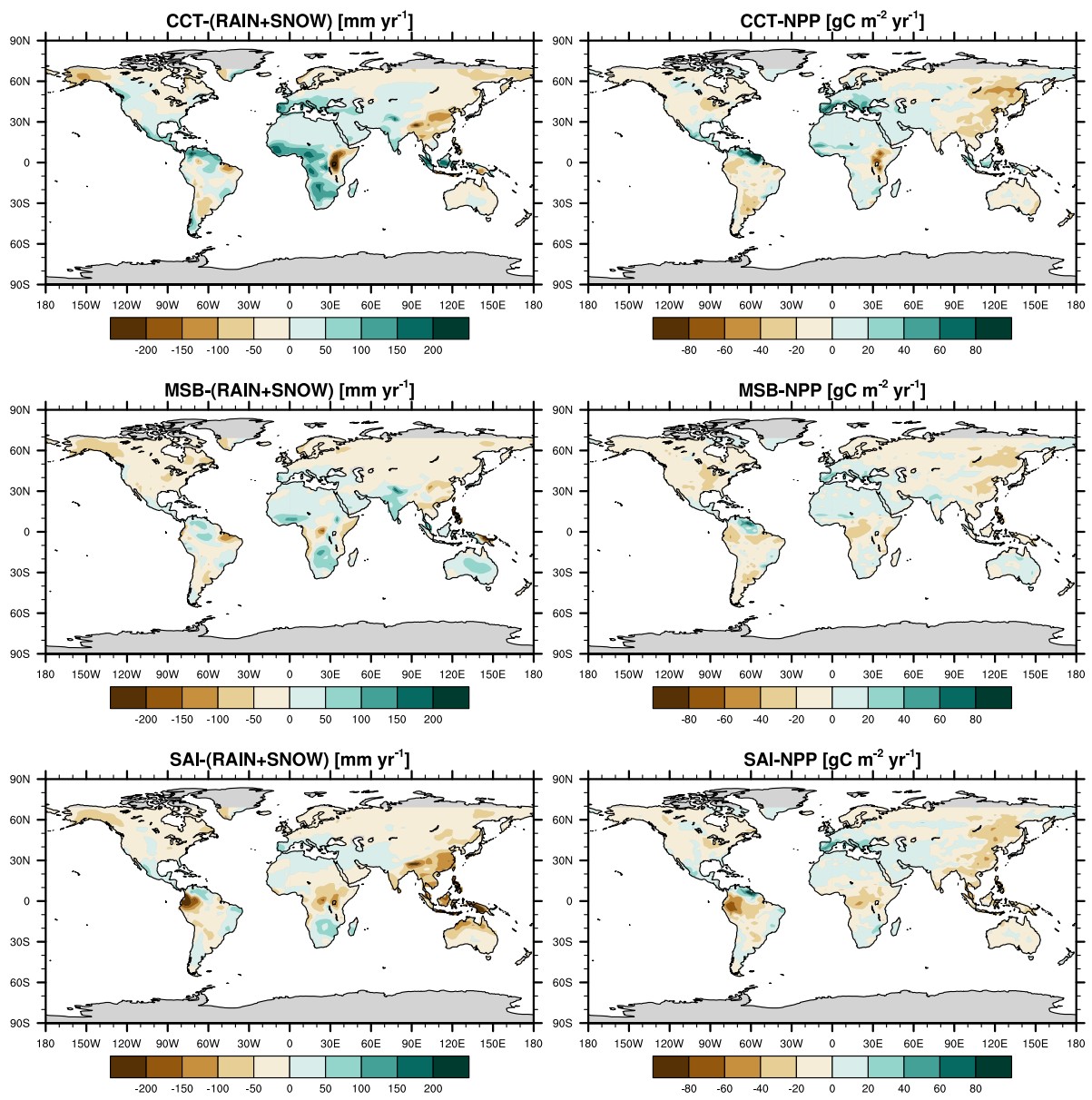

**Figure 2.** The deviation of precipitation and NPP simulated by CCT, MSB, and SAI relative to the baseline RCP8.5 scenario. The values shown here are the mean difference of the 2070-2100 time period and the mean over three ensemble members.

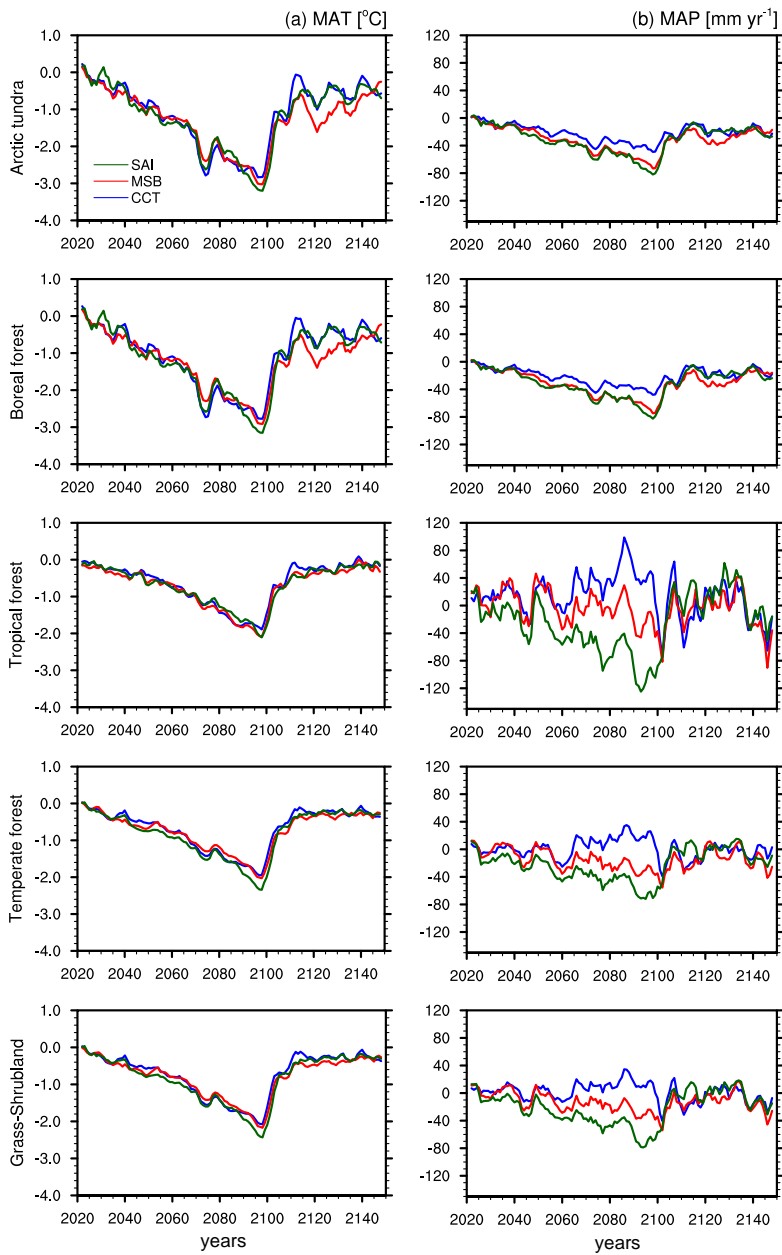

**Figure 3.** Mean annual temperature and precipitation in five different land biomes existing from -60 to 70°N latitude. The changes are relative to the baseline RCP8.5 scenario.

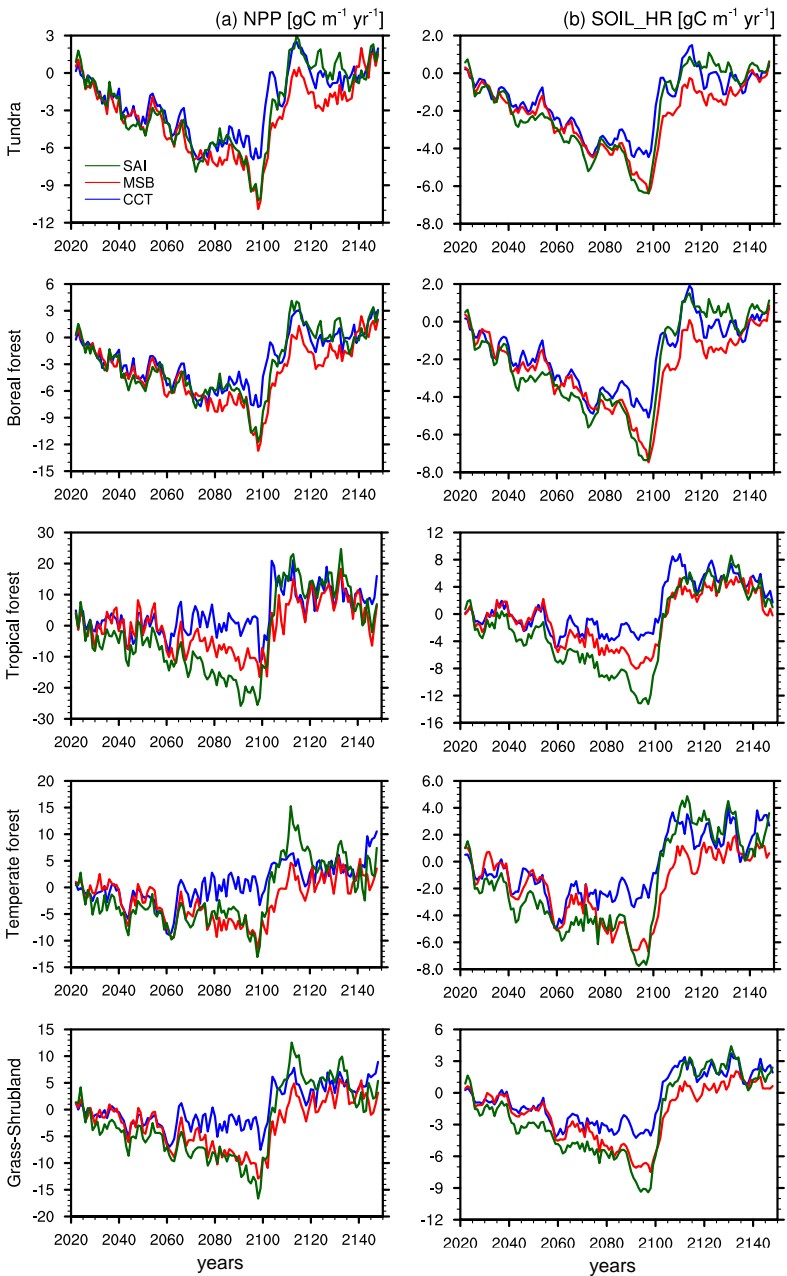

**Figure 4.** The relative difference between the RM to RCP8.5 scenario. The values are mean biome NPP and SOIL_HR in land areas across -60 to 70°N latitude.

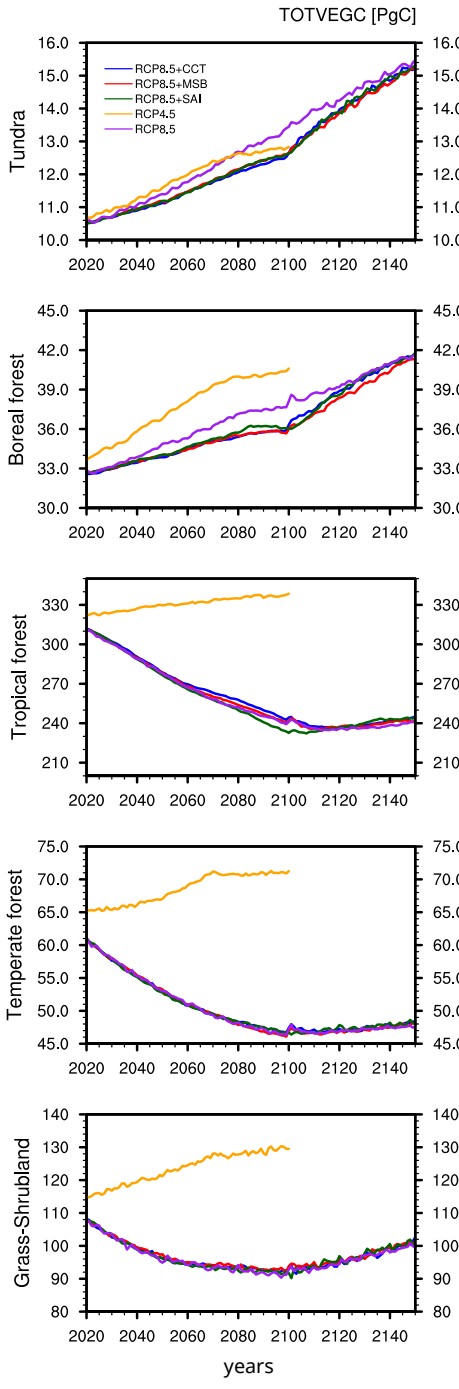

**Figure 5.** Total vegetation carbon storage in five different biomes under simulations of RCP8.5, three RM methods applied on top of RCP8.5 climate forcing (CCT, MSB, and SAI), and RCP4.5 scenarios.