# Peer review of "The response of terrestrial ecosystem carbon cycling under different aerosol-based radiation management geoengineering"

_Earth System Dynamics, 2020_

## Referee Comment (RC1) · Anonymous Referee #1 · 27 Aug 2020

The manuscript "The response of terrestrial ecosystem carbon cycling under different aerosol-based radiation management geoengineering" by Lee et al. investigated the impacts of 3 different aerosol-based radiation management geoengineering methods on land C cycle using the NorESM1-ME earth system model. By comparing the simulations under RCP 8.5, RCP 4.5 and 3 geoengineering scenarios, the authors suggested that different geoengineering methods can result in very different precipitation patterns in tropical forests and finally affect global C budget. Also, the authors suggested a significant impact from $CO_2$ fertilization.

Generally, this manuscript is well structured and written. Although, some analyses are

recommended to further improve the manuscript. Below are my suggestions:

General comments:

1. The authors investigated the spatial patterns of precipitation changes caused by RM applications, but gave only a little information on the spatial patterns of other variables affecting C cycle. As recently suggested by Zhang et al. (2019), vegetation at different latitudes show different sensitivities to aerosol-caused temperature changes. In Line 168-170, the authors indicated stronger cooling effect of RM in tropics than high latitudes. How is this pattern different from the RCP4.5 scenario? It would be interesting to compare and discuss. If possible [optional], to quantify this impact using sensitivity as in Zhang et al. (2019) or a few offline simulations on the land surface model will be able to distinguish the impacts of temperature (probably other factors) and $CO_2$ fertilization and provide more insightful understanding of RM impacts.

2. Similar to the previous point, I would also check if the downward solar radiation at land surface has different patterns among the 3 RM methods, especially in tropical forests, because some studies indicate radiation-limiting vegetation in these regions (e.g. Nemani et al., 2003).

3. The authors suggested that the diffuse radiation is not so important regulating NPP in Line 160. To justify this, the authors need to clarify whether and how the model distinguish diffuse and direct radiation in the Model description section. Also, I recommend the authors to use total downward surface radiation and diffuse radiation fraction rather than direct and diffuse radiation because the previous way can more clearly distinguish the effect of aerosol-caused dimming and increase in diffuse light.

4. It is not clear whether SI1 is the spatial pattern of temporal correlation, or the pattern of spatial correlation calculated in groups of nearby grids? The later correlation might indicate mainly the response of PFT distribution to precipitation regime given the coarse resolution of the model. In the manuscript, to investigate the response of the vegetation to precipitation changes, the temporal correlation (partial correlation to

control temperature and radiation) is more reasonable.

Other comments:

Line 153 "large overall". Also to be consistent in tense. For instance, Line 153 used "is" but Line 155 used "was"

Line 189-191: Not clear, need to be rephrased

Overall, this is a good study and relevant to ESD. I recommend to publish it with the above points addressed.

Mentioned References: Nemani, R. R., Keeling, C. D., Hashimoto, H., Jolly, W. M., Piper, S. C., Tucker, C. J., Myneni, R. B., and Running, S. W.: Climate-driven increases in global terrestrial net primary production from 1982 to 1999, Science, 300, 1560-1563, 2003. Zhang, Y., Goll, D., Bastos, A., Balkanski, Y., Boucher, O., Cescatti, A., Collier, M., Gasser, T., Ghattas, J., Li, L., Piao, S., Viovy, N., Zhu, D., and Ciais, P.: Increased Global Land Carbon Sink Due to Aerosol-Induced Cooling, Global Biogeochemical Cycles, 33, 439-457, 10.1029/2018gb006051, 2019.

---

## Referee Comment (RC2) · Anonymous Referee #2 · 30 Aug 2020

This manuscript studied how vegetation carbon and soil carbon changes under three solar radiation management: sulfate aerosol injection, marine cloud brightening and cirrus cloud thinning. The three SRM strategies result different responses of terrestrial carbon pools. Based on analysis of temperature and precipitation changes, they concluded that precipitation is the factor to determine the terrestrial carbon storage under SRM scenarios comparing with RCP8.5. They also concluded that $CO_2$ concentration and land surface usage play a more important role than climate changes under SRM. There are only a few works on SRM impacts on vegetation and terrestrial carbon cycle, and more research are needed to further understand vegetation and carbon cycle responses to climate changes from SRM. Therefore, this study is filling the gap

in SRM impact study and is the first to compare three SRM methods. However, some improvements are needed in writing and analysis.

General Comments:

This manuscript only analyzed temperature and precipitation as biome average and concluded precipitation is the determine factor for plant NPP. However, the relation between precipitation and NPP may not be cause and effect. (1) other climate factors important to vegetation and soil respiration are ignored. Diffuse and direct radiation have only be looked at as land average. $CO_2$ fertilization effect might be different under different combination of T&P. Comparing to precipitating, soil moisture is more important to plant growth. (2) within one biome, is NPP change homogeneous? What about temperature and precipitation? What is the fundamental phenology to explain the relationship between precipitation and NPP/soil respiration? In addition, please don't repeat the same sentences in abstract, introduction and conclusion.

Specific Comments:

1. Line 2: please change "and at the same time help reach" to "by reaching"

2. Line 4-6: Please change the sentence to something like "Here we assess the changes of ecosystem carbon exchange and storage among different terrestrial biomes under three aerosol based radiation management (RM) methods with the baseline of RCP8.5 using an Earth System Model (NorESM1-ME)."

3. Line 7 and in the whole manuscript: please change "marine sky brightening" to "marine cloud brightening", "MSB" to "MCB"

4. Line 8: please add "reach" before "that of the RCP4.5 scenario"

5. Line 8: please delete "different" after "three". Please change "exhibit" to "show"

6. Line 9: please delete "due to the methodological differences in how the aerosols are applied"

7. Line 10-11: please change the sentence to "Precipitation differences from the three RM methods result large variability in global vegetation carbon uptake and storage"

8. Line 11: add some explanation, why "these consequences should be taken into account in future climate policies"

9. Line 14-17: rephrase the two sentences "We find that changes in vegetation ...in the tropics. Hence..."

10. Line 21: please change "Reaching global climate target of 1.5-2 oC" to "This temperature target"

11. Line 22: please change "but also" to "and also"

12. Line 23-26: please rephrase the two sentences. They are the same as in the abstract.

13. Line 34-35: What is the reason of using RM instead of SRM? SRM is a much more used term in the field. Originally, I thought the reason of using RM is because CCT is included, which is management of longwave radiation instead of solar radiation. But now I am confused.

14. Line 44: please rephrase "unless atmospheric CO2 concentrations are not dealt with during such a RM deployment period"

15. Line 47: please add references: Xia et al., 2016, Yang et al., 2020

Xia, Lili, Alan Robock, Simone Tilmes, and Ryan R. Neely III, 2016: Stratospheric sulfate geoengineering could enhance the terrestrial photosynthesis rate. Atmos. Chem. Phys., 16, 1479-1489, doi:10.5194/acp-16-1479-2016.

Yang et al., (2020) Assessing terrestrial biogeochemical feedbacks in a strategically geoengineered climate, Environmental Research Letters, https://doi.org/10.1088/1748-9326/abacf7

16. Line 53: please correct this sentence "The mechanism how different methods stabilize the climate are slightly different." The mechanism of the three methods are quite different. SAI and MCB are managing shortwave radiation, and CCT is modifying terrestrial radiation.

17. Line 59-60: please clarify the two sentences: is the precipitation the main reason for the different responses in global average and high latitude NPP? That isn't cause and effect. The difference of NPP in global mean and high latitude is due to many reasons, such as regional changes, different combination of temperature/precipitation. In addition, I don't understand why comparing global mean and high latitude? Or maybe the author tries to say there are different NPP responses in climate models? In that case, precipitation is not the main reason. The different land models and how they represent C-N cycle are the main reason.

18. Line 79-89: more introductions on land model: how carbon pools are calculated, how soil carbon and nitrogen is affected by climate changes, how photosynthesis is represented... Also, since land model is adopted from CLM4, if comparing with CLM5, what are the advantages and disadvantages?

19. Line 104: Tilmes et al. (2015) is to reduce radiative forcing from RCP6.0 to RCP4.5. To clarify, maybe add something like "based on the approach of Tilmes et al., (2015), although different reference cases are used."

20. Figure 1 and Line 137-141: It is better to show the temperature response or the radiative forcing change TOA first. Then describe details of direct and diffuse radiation. CO2 concentration change should be the last, after all interactions with the terrestrial system and ocean.

21. Figure 1: (1) please use the full names for subtitles (and please do this for all figures); (2) CO2 in subtitle should be $CO_2$; (3) please indicate what is "-60 to 70°N"; (4) please change the line thickness of plots' boundaries thinner (please do this for all figures); (4) please show the radiation at TOA first; (5) why are temperature reduction in

three methods similar, but soil respiration changes are different? (6) Why does RCP4.5 show much lower soil respiration than three RM scenarios?

22. Line 146: please explain why CCT has precipitation higher than RCP8.5 here instead of saying the reason is in Muri et al., 2018. You can explain it here, and then cite Muri et al., 2018 to support.

23. Line 148: please cite Figure 2 for precipitation pattern changes.

24. Line 153: please change "largeo" to "large". The increase is compared to which period?

25. Line 156-159: please break this long sentence into two or three sentences.

26. Line 160: please add reference Xia et al., 2016. Did you look at photosynthesis changes under those scenarios? Although there is no significant change in NPP, there might be changes in photosynthesis rate and GPP, but respiration changes may play an important role in NPP.

27. Line 163-166: the description of precipitation pattern change under three scenarios are not accurate. Middle latitude over Northern Hemisphere are important in term of NPP and agriculture. And all methods show precipitation reduction over east Asia.

28. Line 167: please rephrase this sentence.

29. Line 171-172: "In all biomes, SAI application results in the largest decrease in precipitation followed by MSB, and CCT relative to RCP8.5 scenario." Is not accurate, since CCT results more precipitation than in RCP8.5 in some biomes.

30. Line 179: precipitation in SAI does get less than RCP4.5 even averaging only land (Figure 1).

31. Line 181-186: it might be better to describe the overall pattern of NPP change in three scenarios (reduction over the high latitude and spread responses over the mid-low latitudes), and explain why, then get into details of how precipitation difference

might control NPP responses over sub-tropic and tropic regions.

32. Line 181-182: why showing the correlation between precipitation and NPP? This cannot support NPP changes in RM scenarios are due to precipitation changes.

33. Supplementary information 3: please change figure caption. This is average of biomes not -60 to 70°N land. Please use RCP8.5, RCP4.5 in the legend instead of R8.5 and R4.5 (also in all other figures in supplementary materials).

34. Line 202-206, please rephrase this sentence.

35. Line 209-213: Based on Figure 3 and 4, NPP and soil respiration show similar changes as precipitation in temperate forest and grass-shrubland biomes as well. In Figure 5, temperate forest and grass-shrubland should show similar pattern as NPP under three RM scenarios (as they all use the same land surface data as in RCP8.5). The changes are not obvious in Figure 5 is because of the scale and the accumulative C.

36. Figure 4, please use the same scale.

37. Figure 4: unit is wrong, should be gC m-2 yr-1

38. Figure 4: why does soil respiration show similar patterns as in precipitation? Isn't soil respiration strongly affected by temperature? How does soil moisture change?

39. Figure 4: why does NPP under RCP8.5-SAI show a jump after termination, especially in temperate forest and grass-shrubland.

40. Line 214: "This likely due to increased respiration rate overshadowing the increase in NPP", this sentence is incorrect. NPP is GPP minus plant respiration already. If you mean soil respiration, then this respiration has nothing to do with vegetation C.

41. Supplementary information 5: how did soil carbon anomalies calculated? Geo scenarios minus transient RCP8.5? What is the pattern of soil carbon change under RCP8.5?

42. Line 222: please compare with the work from Yang et al., (2020) Assessing terrestrial biogeochemical feedbacks in a strategically geoengineered climate, Environmental Research Letters, https://doi.org/10.1088/1748-9326/abacf7

43. Line 231: what does "the rate of carbon uptake" mean here? NPP or net ecosystem carbon flux?

44. Line 235: could you please estimate how much $CO_2$ concentration change will be based on this 170 PgC difference. Under RCP4.5 and RCP8.5, the anthropogenic $CO_2$ emission difference will cause different $CO_2$ concentration. But what will be $CO_2$ concentration different resulting from the land surface usage difference?

45. Line 239: NPP changes under three RM scenarios do have similar patterns with precipitation changes, but that doesn't mean precipitation change is the reason of NPP changes. Soil respiration is also showing similar pattern. Could the NPP changes due to nitrogen limitation from organic matter decomposition?

46. Line 240-242: This might be incorrect: (1) studies show that cooling over tropics has positive impact on crops comparing with global warming; (2) agriculture will have irrigation, which will be different than natural vegetation.

47. Line 251-252: why is there an increase of soil carbon under RCP4.5? what about RCP8.5?

---

## Author Comment (AC1) · 9 Feb 2021

We appreciate the reviewer's support and comments. The suggestions were very helpful and we revised the manuscript according to the suggestions made by the reviewer.

General comments: 1. The authors investigated the spatial patterns of precipitation changes caused by RM applications, but gave only a little information on the spatial patterns of other variables affecting C cycle. As recently suggested by Zhang et al. (2019), vegetation at different latitudes show different sensitivities to aerosol-caused temperature changes. In Line 168-170, the authors indicated stronger cooling effect of RM in tropics than high latitudes. How is this pattern different from the RCP4.5 scenario? It would be interesting to compare and discuss. If possible [optional], to quantify this impact using sensitivity as in Zhang et al. (2019) or a few offline simulations on the land surface model will be able to distinguish the impacts of temperature (probably other factors) and CO2 fertilization and provide more insightful understanding of RM impacts. 2. Similar to the previous point, I would also check if the downward solar radiation at land surface has different patterns among the 3 RM methods, especially in tropical forests, because some studies indicate radiation-limiting vegetation in these regions (e.g. Nemani et al., 2003). 3. The authors suggested that the diffuse radiation is not so important regulating NPP in Line 160. To justify this, the authors need to clarify whether and how the model distinguish diffuse and direct radiation in the Model description section. Also, I recommend the authors to use total downward surface radiation and diffuse radiation fraction rather than direct and diffuse radiation because the previous way can more clearly distinguish the effect of aerosol-caused dimming and increase in diffuse light.

»The first three comments are related and we address them together here. While investigating the reviewer's comments, we recognized that we made a serious under-investigation of radiation changes in our manuscript. We have investigated changes in radiation and its effects more in depth. We now added additional paragraph describing the patterns of direct and diffuse radiation change (see sections 2.1-2.3).

We realize that the statement we previously made 'the diffuse radiation is not so important regulating NPP in Line 160' was incorrect and now revised it in the main text. In the CLM, both direct and diffuse radiation is used in the computation of photosynthesis and as diffuse radiation is used in both sunlit and shaded leaves, it is indeed more efficient in the overall photosynthetic process as pointed out by several studies. We greatly appreciate the reviewer by pointing out this mistake and giving us the opportunity to correct it.

The reviewer suggested us to investigate sensitivity of photosynthesis using sensitivity test shown in Zhang et al. (2019). We investigated Zhang et al. (2019) and the method

used in this study and decided to keep the analysis as correlation instead of sensitivity. We decided that correlation is a more effective metric to show in our study as it is a more direct comparison of changes between the two variables. Since our study deals with different model experiments using the same model, correlation is simpler and direct method to show how different environmental changes affect photosynthesis. The correlation plot in our study does exhibit some regions to be sensitive to changes in temperature and radiation. It is important to note here that the main goal of our study is to compare how different RM applications affect global biome NPP and C storage. Since we are using the same model, looking into sensitivity in different regions is slightly out of the scope of the study as it does not influence a great deal in comparing different methods of RM and how the different methods manifest in the NPP.

After investigating the changes in direct radiation, diffuse radiation, temperature, and precipitation, we are still convinced that changes in precipitation is the largest driver of changes in NPP under the three RM methods applied. The increases in diffuse radiation may positively influence the rate of photosynthesis, however, the large decrease in precipitation under some of the RM methods applied in our study has greater impact on photosynthesis. We think that this is still a very important investigation made under the suggestion of the reviewer and we truly appreciate this comment. As admitted by the authors in Zhang et al. (2019) study, some of these environmental variables covary. For instance, it is common that where direct radiation is reduced precipitation increases due to increased cloud formation. This is very difficult to partition and may need separate simulations to quantify it. The sensitivity change in various regions under RM is still an interesting topic, but this could better be addressed using experiments conducted using different models. We think this could be an interesting research topic for another study but is a bit out of the scope for this paper that focuses on comparing the impacts of applying three different RM methods.

4. It is not clear whether SI1 is the spatial pattern of temporal correlation, or the pattern of spatial correlation calculated in groups of nearby grids? The later correlation

might indicate mainly the response of PFT distribution to precipitation regime given the coarse resolution of the model. In the manuscript, to investigate the response of the vegetation to precipitation changes, the temporal correlation (partial correlation to control temperature and radiation) is more reasonable.

»We believe the reviewer is addressing SI2 (now SI3) instead of SI1 in this comment. The results in SI2 is simply drawn from a distribution of PFT used in our model. Perhaps the confusion comes from the title of the figure. We appreciate the reviewer's comment for us to clarify this. We have now revised the title to 'Spatial pattern of temporal correlation between the simulated results between RCP8.5 and the RM methods based on RCP8.5 scenario. The variables shown are FVR-NPP, TSA-NPP, precipitation-NPP in each grid of the model mean over the 2070-2100 period.'

Other comments: Line 153 "large overall". Also to be consistent in tense. For instance, Line 153 used "is" but Line 155 used "was"

»Revised to be consistent and as suggested.

Line 189-191: Not clear, need to be rephrased

»Revised.

---

## Author Comment (AC2) · 9 Feb 2021

We appreciate the Reviewer#2's thorough comments. We believe these comments are very helpful in improving the overall quality of the manuscript and we have accepted and addressed most everything the reviewer commented.

Precipitation and NPP

The major concerns the Reviewer#2 raises is how we make connections between climatic factors and vegetation growth. While we agree that soil moisture rather than precipitation is a more accurate parameter that affects photosynthesis, we still argue

that the main objective of this study is how climatic factors affect terrestrial carbon storage. Similar to how soil temperature is a function of air temperature, soil moisture has a direct correlation to precipitation. This is a widely accepted relationship that is widely used to argue how climatic factors affect the functioning of photosynthesis as well as microbial respiration. Therefore, we do not feel the need to make this as part of the underlying argument of our analyses.

Vegetation phenology

As we show in the Supplementary Information 4, we do not see much change in seasonality in NPP and TLAI, thus did not mention much about how RM affect phenology.

Below are our response to the specific comments.

1. Line 2: please change "and at the same time help reach" to "by reaching"

»Revised as suggested.

2. Line 4-6: Please change the sentence to something like "Here we assess the changes of ecosystem carbon exchange and storage among different terrestrial biomes under three aerosol based radiation management (RM) methods with the baseline of RCP8.5 using an Earth System Model (NorESM1-ME)."

»Revised as suggested

3. Line 7 and in the whole manuscript: please change "marine sky brightening" to "marine cloud brightening", "MSB" to "MCB"

»In order to clarify this, we have added a new paragraph describing the justification of the terms used in our study. Please see pg2.

4. Line 8: please add "reach" before "that of the RCP4.5 scenario"

»Revised as suggested.

5. Line 8: please delete "different" after "three". Please change "exhibit" to "show"

»Revised as suggested.

6. Line 9: please delete "due to the methodological differences in how the aerosols are applied"

»Revised as suggested.

7. Line 10-11: please change the sentence to "Precipitation differences from the three RM methods result large variability in global vegetation carbon uptake and storage"

»Revised as suggested.

8. Line 11: add some explanation, why "these consequences should be taken into account in future climate policies"

»This part is revised to 'Our findings show that there are unforeseen regional consequences under geoengineering and these consequences should be taken into account in future climate policies as they have substantial impact on terrestrial ecosystems.'

9. Line 14-17: rephrase the two sentences "We find that changes in vegetation . . .in the tropics. Hence. . ."

»Revised as suggested.

10. Line 21: please change "Reaching global climate target of 1.5-2 oC" to "This temperature target"

»Revised as suggested.

11. Line 22: please change "but also" to "and also"

»Revised as suggested.

12. Line 23-26: please rephrase the two sentences. They are the same as in the abstract.

»Revised as suggested.

13. Line 34-35: What is the reason of using RM instead of SRM? SRM is a much more used term in the field. Originally, I thought the reason of using RM is because CCT is included, which is management of longwave radiation instead of solar radiation. But now I am confused.

»We use the term RM since we are also looking at cirrus cloud thinning which works to increase longwave radiation. We have included an explanation of this on p. 2.

14. Line 44: please rephrase "unless atmospheric CO2 concentrations are not dealt with during such a RM deployment period"

»Revised as suggested.

15. Line 47: please add references: Xia et al., 2016, Yang et al., 2020 Xia, Lili, Alan Robock, Simone Tilmes, and Ryan R. Neely III, 2016: Stratospheric sulfate geoengineering could enhance the terrestrial photosynthesis rate. Atmos. Chem. Phys., 16, 1479-1489, doi:10.5194/acp-16-1479-2016. Yang et al., (2020) Assessing terrestrial biogeochemical feedbacks in a strategically geoengineered climate, Environmental Research Letters, https://doi.org/10.1088/1748-9326/abacf7

»Added as suggested.

16. Line 53: please correct this sentence "The mechanism how different methods stabilize the climate are slightly different." The mechanism of the three methods are quite different. SAI and MCB are managing shortwave radiation, and CCT is modifying terrestrial radiation.

»Revised as suggested.

17. Line 59-60: please clarify the two sentences: is the precipitation the main reason for the different responses in global average and high latitude NPP? That isn't cause and effect. The difference of NPP in global mean and high latitude is due to many reasons, such as regional changes, different combination of temperature/precipitation. In addition, I don't understand why comparing global mean and high latitude? Or maybe

the author tries to say there are different NPP responses in climate models? In that case, precipitation is not the main reason. The different land models and how they represent C-N cycle are the main reason.

»We address this above in our main response.

18. Line 79-89: more introductions on land model: how carbon pools are calculated, how soil carbon and nitrogen is affected by climate changes, how photosynthesis is represented. . . Also, since land model is adopted from CLM4, if comparing with CLM5, what are the advantages and disadvantages?

»Additional information added to describe C pools. We disagree with the reviewer on the latter point. Since only one model is used in our study, we do not see the reason to make comparisons. We believe comparing the performance and parameterization between CLM4 and CLM5 is beyond the scope of this study.

19. Line 104: Tilmes et al. (2015) is to reduce radiative forcing from RCP6.0 to RCP4.5. To clarify, maybe add something like "based on the approach of Tilmes et al., (2015), although different reference cases are used."

»Revised as suggested.

20. Figure 1 and Line 137-141: It is better to show the temperature response or the radiative forcing change TOA first. Then describe details of direct and diffuse radiation. CO2 concentration change should be the last, after all interactions with the terrestrial system and ocean.

»We have revised the figure based on the reviewer's suggestions. Note that we also revised the figure legend.

21. Figure 1: (1) please use the full names for subtitles (and please do this for all figures); (2) CO2 in subtitle should be CO2; (3) please indicate what is "-60 to 70âŮęN"; (4) please change the line thickness of plots' boundaries thinner (please do this for all figures); (4) please show the radiation at TOA first; (5) why are temperature reduction in

three methods similar, but soil respiration changes are different? (6) Why does RCP4.5 show much lower soil respiration than three RM scenarios?

»We have revised the Figure 1 based on the reviewer's suggestions. In this study we choose to use the incoming direct and diffuse radiation components for our analysis. The TOA variables are not part of our analysis given that the surface radiation variables are the most important for photosynthesis and NPP. Different precipitation reductions in three methods leads to different soil respiration changes. Also the lower temperature values in RCP4.5 cause lower respiration rates.

22. Line 146: please explain why CCT has precipitation higher than RCP8.5 here instead of saying the reason is in Muri et al., 2018. You can explain it here, and then cite Muri et al., 2018 to support.

»Thank you for this comment. This is a fair point. We have now added the following to the manuscript: "CCT has been shown to lead to an increase in precipitation in previous studies (Kristjansson et al., 2015, Jackson et al., 2016, Muri et al., 2018), where the radiative cooling of the troposphere increases the latent heat flux at the surface and hence the precipitation rates".

23. Line 148: please cite Figure 2 for precipitation pattern changes.

»Revised as suggested.

24. Line 153: please change "largeo" to "large". The increase is compared to which period?

»Revised as suggested.

25. Line 156-159: please break this long sentence into two or three sentences.

»This part is now separated into three sentences to help the readers.

26. Line 160: please add reference Xia et al., 2016. Did you look at photosynthesis changes under those scenarios? Although there is no significant change in NPP, there

might be changes in photosynthesis rate and GPP, but respiration changes may play an important role in NPP.

»Now we refer to Xia et al. 2016 as suggested. We did not because we were primarily interested in carbon storage. Also, both processes being biological process, the rate of photosynthesis would parallel photorespiration.

27. Line 163-166: the description of precipitation pattern change under three scenarios are not accurate. Middle latitude over Northern Hemisphere are important in term of NPP and agriculture. And all methods show precipitation reduction over east Asia.

»Revised as suggested.

28. Line 167: please rephrase this sentence.

»Revised as suggested.

29. Line 171-172: "In all biomes, SAI application results in the largest decrease in precipitation followed by MSB, and CCT relative to RCP8.5 scenario." Is not accurate, since CCT results more precipitation than in RCP8.5 in some biomes.

»Revised as suggested.

30. Line 179: precipitation in SAI does get less than RCP4.5 even averaging only land (Figure 1).

»This description is revised now.

31. Line 181-186: it might be better to describe the overall pattern of NPP change in three scenarios (reduction over the high latitude and spread responses over the mid-low latitudes), and explain why, then get into details of how precipitation difference might control NPP responses over sub-tropic and tropic regions.

»We appreciate this comment. Based on the recommendation from the reviewer, we now added the following description under 'Regional differences in temperature and

precipitation' subsection.

"There are common spatial patterns in NPP decrease at north western part of Amazonia, equatorial Africa, and eastern Asia in the three RM experiments (Figure 2). But overall, the large increase in NPP at Europe and equatorial South America particularly CCT experiment compensates the decreases elsewhere, hence creating a general lack of deviation from RCP8.5 scenario (see Figure 1). It is clear from the comparison shown in precipitation (Figure 2) that the NPP changes are most affected by the spatial changes in precipitation."

32. Line 181-182: why showing the correlation between precipitation and NPP? This cannot support NPP changes in RM scenarios are due to precipitation changes.

»Revised as suggested.

33. Supplementary information 3: please change figure caption. This is average of biomes not -60 to 70 âŮę N land. Please use RCP8.5, RCP4.5 in the legend instead of R8.5 and R4.5 (also in all other figures in supplementary materials).

»Corrected as suggested.

34. Line 202-206, please rephrase this sentence.

»Revised as suggested.

35. Line 209-213: Based on Figure 3 and 4, NPP and soil respiration show similar changes as precipitation in temperate forest and grass-shrubland biomes as well. In Figure 5, temperate forest and grass-shrubland should show similar pattern as NPP under three RM scenarios (as they all use the same land surface data as in RCP8.5). The changes are not obvious in Figure 5 is because of the scale and the accumulative C.

»This was done on purpose to exhibit the vast differences in C accumulation when comparing to RCP4.5 scenario.

36. Figure 4, please use the same scale.

»We disagree. Different scales are used to most effectively show the differences across the three RM methods.

37. Figure 4: unit is wrong, should be gC m-2 yr-1

»Revised as suggested.

38. Figure 4: why does soil respiration show similar patterns as in precipitation? Isn't soil respiration strongly affected by temperature? How does soil moisture change?

»This was addressed in the main response above.

39. Figure 4: why does NPP under RCP8.5-SAI show a jump after termination, especially in temperate forest and grass-shrubland.

»In both temperate forest and grass-shrubland, SAI simulates the largest decline in precip, followed by largest recovery (Fig. 3) after termination. This may have caused a sudden jump in NPP right after termination. The other potential reason is light. SAI also simulates largest loss and recovery of incoming radiation (direct+diffuse) before and after termination, respectively.

40. Line 214: "This likely due to increased respiration rate overshadowing the increase in NPP", this sentence is incorrect. NPP is GPP minus plant respiration already. If you mean soil respiration, then this respiration has nothing to do with vegetation C.

»This sentence is deleted to remove confusion.

41. Supplementary information 5: how did soil carbon anomalies calculated? Geo scenarios minus transient RCP8.5? What is the pattern of soil carbon change under RCP8.5?

»Revised to 'The relative difference in total vegetation and soil carbon between the RM to RCP8.5 scenario.' The overall pattern can be inferred from SI6.

42. Line 222: please compare with the work from Yang et al., (2020) Assessing terrestrial biogeochemical feedbacks in a strategically geoengineered climate, Environmental Research Letters, https://doi.org/10.1088/1748-9326/abacf7

»We really appreciate this suggestion. Now revised to 'The likely accumulation of soil carbon under RM application may be viewed as one of the positive effects of geoengineering, which was supported by a recent multi-model comparison study (Yang et al. 2020).'

43. Line 231: what does "the rate of carbon uptake" mean here? NPP or net ecosystem carbon flux?

»This is NPP and we've corrected this in the text.

44. Line 235: could you please estimate how much $CO_2$ concentration change will be based on this 170 PgC difference. Under RCP4.5 and RCP8.5, the anthropogenic $CO_2$ emission difference will cause different $CO_2$ concentration. But what will be $CO_2$ concentration different resulting from the land surface usage difference?

»170 PgC bound in vegetation carbon would translate to 80 ppm difference in atmospheric concentration, if there were no other changes in the system. We added this to the text.

45. Line 239: NPP changes under three RM scenarios do have similar patterns with precipitation changes, but that doesn't mean precipitation change is the reason of NPP changes. Soil respiration is also showing similar pattern. Could the NPP changes due to nitrogen limitation from organic matter decomposition?

»Soil respiration, being a biological process, is also affected by the precipitation changes similar to NPP. Therefore, similar patterns are observed between NPP and soil respiration. Nitrogen limitation can affect overall biological response, however, this study focuses on the effects of environmental changes on carbon balance given that the same land model version was used in all three RM experiments. This could be

of an interesting research question to address using the state-of-the-art land model versions with better representation of nitrogen cycling.

46. Line 240-242: This might be incorrect: (1) studies show that cooling over tropics has positive impact on crops comparing with global warming; (2) agriculture will have irrigation, which will be different than natural vegetation.

»We agree that considering only cooling impact, a previous study has indicated SRM will induce positive impact on yield production (e.g., Pongratz et al., 2012; https://www.nature.com/articles/nclimate1373). Precipitation response is regionally more complex than temperature, and the same study states that SRM-induced reduction in Asian monsoon lead to few percent reduction of crop yield and only partly offset by the cooling effect. And we also agree that impact on irrigated and non-irrigated crops will be unequal. We have therefore revises the sentence from "This implies that RM may have negative effects on food production globally . . ." to "Therefore, considering the changes (i.e., reduction) in precipitation alone, RM may have negative effects on non-irrigated crops or food production globally. Nevertheless, the effect of $CO_2$ fertilization effect in the future are suggested to compensate the deleterious impacts of both RM-induced temperature and precipitation changes (Pongratz et al., 2012; Xia et al., 2014)."

47. Line 251-252: why is there an increase of soil carbon under RCP4.5? what about RCP8.5?

»We now explain this in more detail in the text.